# Text mining occupations from the mental health electronic health record: a natural language processing approach using records from the Clinical Record Interactive Search (CRIS) platform in south London, UK

Natasha Chilman ,[1] Xingyi Song ,[2] Angus Roberts ,[1] Esther Tolani ,[1] Robert Stewart ,[1,3] Zoe Chui ,[1] Karen Birnie ,[1,4] Lisa Harber-Aschan ,[1] Billy Gazard ,[1] David Chandran ,[3] Jyoti Sanyal,[3] Stephani Hatch ,[1,5] Anna Kolliakou ,[1] Jayati Das-Munshi [1,3,5]

► Prepublication history and supplemental material for this paper is available online. To view these files, please visit the journal online (http://dx.doi.org/10.1136/bmjopen-2020-042274).

AK and JD-M are joint senior authors.

For numbered affiliations see end of article.

**Correspondence to**
Natasha Chilman;
natasha.chilman@kcl.ac.uk

## ABSTRACT

**Objectives** We set out to develop, evaluate and implement a novel application using natural language processing to text mine occupations from the free-text of psychiatric clinical notes.

**Design** Development and validation of a natural language processing application using General Architecture for Text Engineering software to extract occupations from de-identified clinical records.

**Setting and participants** Electronic health records from a large secondary mental healthcare provider in south London, accessed through the Clinical Record Interactive Search platform. The text mining application was run over the free-text fields in the electronic health records of 341 720 patients (all aged ≥16 years).

**Outcomes** Precision and recall estimates of the application performance; occupation retrieval using the application compared with structured fields; most common patient occupations; and analysis of key sociodemographic and clinical indicators for occupation recording.

**Results** Using the structured fields alone, only 14% of patients had occupation recorded. By implementing the text mining application in addition to the structured fields, occupations were identified in 57% of patients. The application performed on gold-standard human-annotated clinical text at a precision level of 0.79 and recall level of 0.77. The most common patient occupations recorded were 'student' and 'unemployed'. Patients with more service contact were more likely to have an occupation recorded, as were patients of a male gender, older age and those living in areas of lower deprivation.

**Conclusion** This is the first time a natural language processing application has been used to successfully derive patient-level occupations from the free-text of electronic mental health records, performing with good levels of precision and recall, and applied at scale. This may be used to inform clinical studies relating to the broader social determinants of health using electronic health records.

### Strengths and limitations of this study

► The application was developed on a sizeable corpus of training and test data from a large routine dataset, which was applied at scale over the record, providing us with insights into the occupations of patients using secondary mental health services.

► The application was thoroughly evaluated using gold-standard and cross-checking strategies.

► The application was developed and tested in a single site electronic health record system in the UK—the application will require validation on other similar systems before using them.

► The application does not identify the temporality of occupations; it is unclear whether the extracted occupations are currently or previously held by the patient.

► The application cannot yet identify where a patient holds a health or social care occupation as these occupations could not be ascertained with confidence.

## INTRODUCTION

Occupation and mental illness are highly inter-related. There are long-standing concerns that unemployment rates are considerably higher for people with mental illness,[1 2] and work participation has been described as among the most important factors for recovery by clinicians and service users alike.[3 4] People with mental illnesses may also undertake precarious, poorly paid work which could have further negative impacts on mental health.[5] Moreover, occupation is a fundamental individual-level indicator of socioeconomic position as it is predictive of material resources and is indicative of wider

class interactions.[6] Recent systematic reviews have called for large and detailed longitudinal studies to investigate predictors of occupational functioning, and to examine how and when occupation is associated with clinical outcomes in mental health cohorts, as this is currently poorly understood.[7 8]

Research using electronic health records (EHRs) allows for the large-scale collection of sociodemographic and clinical information which would otherwise be logistically challenging to collect using traditional epidemiological approaches.[9] However, EHR research has major limitations including that information relating to occupation is either not recorded routinely or is poorly captured within standard EHR systems.[10] As there are no existing methods, to our knowledge, to reliably extract occupations from the psychiatric EHR, this is a problematic barrier for desirable research where occupation is an indicator of socioeconomic status and in research examining the relationships between occupation, mental illness and recovery.

Patient information can be recorded in the structured fields of the EHR, where the clinician records categorical or numerical data. In many psychiatric EHR systems, patient information is recorded in narrative text sections of the record, known as the 'free-text' fields, for example in notes describing patient contact.[11] Information recorded in this way is harder to extract. Clinicians may only record the patient's occupation in such free-text fields and not the structured fields, making it more complicated, time-consuming and labour intensive to identify the patient's occupation.[10] Natural language processing (NLP) methods have the potential to overcome this obstacle by applying algorithms to extract relevant textual information. NLP methods have previously been used successfully for text mining from mental health EHRs, for example, to identify smoking status and symptoms of severe mental illness,[12–16] and other types of clinical records.[17 18] NLP methods are also being applied in large-scale industrial and occupational research.[19–21]

This paper traces the development of a novel application using NLP methods to extract patient occupations from the free-text of EHRs from a large mental health Trust in south London, UK. We then provide profile information on the most frequently extracted occupations for patients using secondary mental health services, and clinical and sociodemographic factors associated with recorded occupation data compared with missing occupation data.

## MATERIALS AND METHODS
### Setting
Data for the development of the application were obtained from the South London and Maudsley (SLaM) Biomedical Research Centre (BRC) case register: a repository of de-identified clinical data from the EHRs of individuals receiving care from SLaM secondary mental health services. SLaM covers a socially and ethnically diverse inner-city area of approximately 1.3 million people.[22] The register contains over 350 000 de-identified patient records which are available for research purposes through the Clinical Record Interactive Search (CRIS) platform. CRIS was developed at SLaM in 2008 and similar resources have subsequently been implemented at several other mental health Trusts in the UK. The present application was developed over the years 2017–2019 and was implemented in January 2020.

### Datasets
Figure 1 describes how the CRIS-derived dataset was used for cycles of application development and evaluation, and summarises the key steps taken. Age restrictions were implemented throughout document selection: free-text documents were only extracted where the patient was aged 16 years and above at time of document extraction. There were no date restrictions. Free-text documents were retrieved from several different sections in this EHR, for example, sections for clinical risk assessments and separate sections for discharge summaries. Further detail on the types of documents used at each stage of application development can be found in online supplemental file 1.

### Developing, evaluating and implementing the application
#### Manually annotating occupation in the free-text
Personal history sections of psychiatric assessments typically describe the patient's occupation, as well as education and family history. Personal history sections of documents were extracted from the free-text fields of records at the document level using an NLP application (precision=0.78, recall=0.88) developed by DC (N=67 383). Typically these extracts were derived from documents of the 'attachments' type, which is a word-processed document such as a letter to or from the patient's primary care physician; and 'events', which are short pieces of text used to record some detail of a clinical encounter.

Occupations were identified in personal history documents by an interdisciplinary team of trained researchers, including clinicians, bioinformaticians and mental health researchers. In common with the NLP community, we refer to this task of marking mentions of occupation text as annotation. A set of occupation annotation guidelines was developed through an iterative process of manual annotation practice, team discussions and agreed annotation rulemaking (online supplemental file 2). These guidelines specified when and how an occupation should be identified, annotated and extracted from the text. An occupation annotation was defined as having two parts. First, the *occupation* itself was annotated. This could be an occupation title, for example, a 'builder'; or an occupation description, for example, 'construction'. Second, the occupation *relation* was specified: who the occupation belongs to, for example, the patient or their family member. Temporality, including when or how long a patient has held an occupation, was not annotated as the text often did not state this consistently. In total, 600 personal history documents were manually

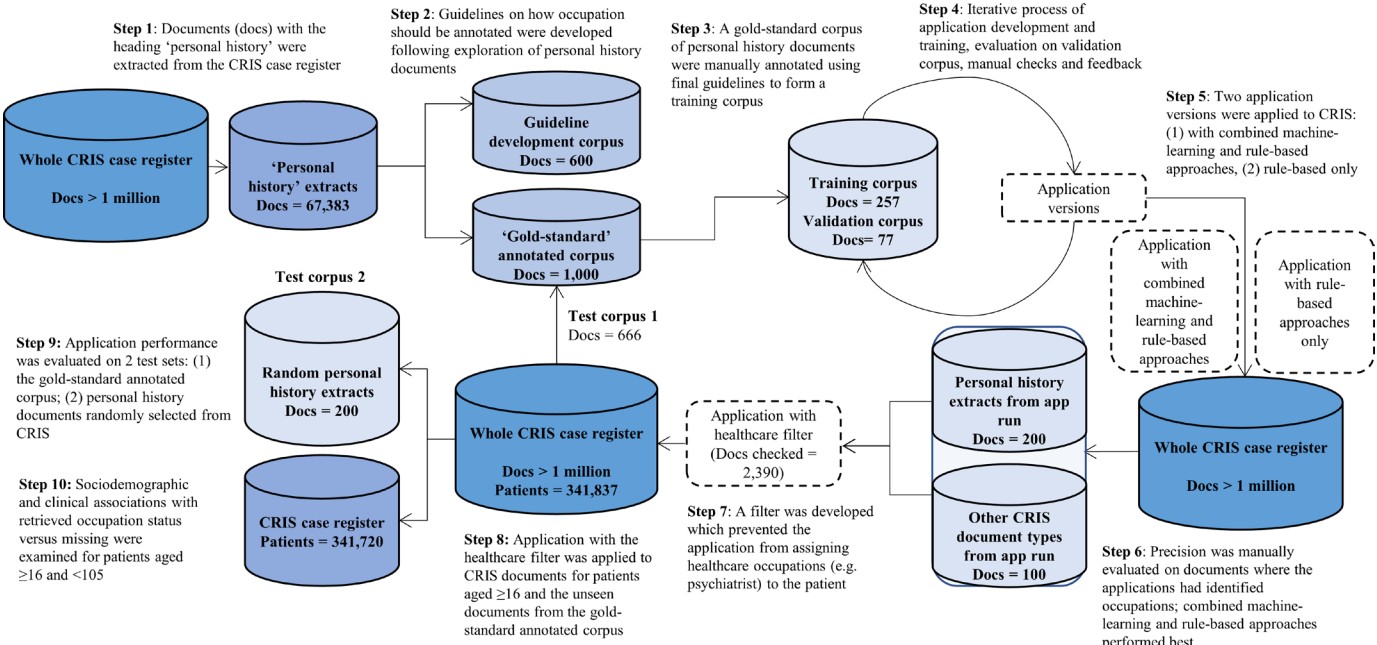

**Figure 1** A step-by-step illustration of the methods used for the occupation application development and evaluation, with the number and types of documents used at each step. CRIS, Clinical Record Interactive Search.

annotated to practise annotating occupation from text and develop the annotation guidelines (ET, AK, SM, KB, ZC, AR). Once the guidelines were developed, a set of 1000 personal history documents were manually annotated on the General Architecture for Text Engineering (GATE) platform[23] using the guidelines to create a gold standard, where 200 were double annotated to evaluate inter-annotator reliability.

## Application development

Out of the 1000 gold-standard annotated personal history documents, 334 documents were reserved for application development. The application was developed by XS on the GATE platform,[23] a widely used NLP framework with over 40 000 downloads per version and a history of use in the UK National Health Service (NHS), among other sectors.[17] The application was trained on 257 of the gold-standard annotated documents. To check the performance of the application throughout development, precision and recall metrics were estimated using a customised performance tool developed by XS on GATE on a validation set of 77 gold-standard annotated documents, with a total of 405 occupation annotations. Precision was the proportion of occupations correctly annotated, to all occupations annotated (whether correct or incorrect). Recall was the proportion of occupations correctly annotated, to all occupations that could have been correctly annotated. The application outputs were manually checked by the Clinical Informatics Interface and Network Lead at the National Institute for Health Research BRC (AK). Any problems identified were addressed in each version of the application. An iterative process of application development, training, evaluation of performance using GATE and manual checks was

repeated 10 times, at which point the application reached a good level of performance on the validation set.

## Machine-learning approach testing

Two early versions of the application were developed for testing over unannotated documents in the CRIS case register: one version used combined machine-learning and rule-based approaches, and the second version used rule-based approaches only. This was due to a concern that the application had therein been developed on limited training data, and the trained model may not generalise well on free-text other than personal history documents, which could lead to a loss in precision when implemented over the EHR. Specifically, the machine-learning approaches involved a trained conditional random field classifier to identify occupation mentions in the text, and a support-vector machine-based classifier to identify the occupation relation. Figure 2 illustrates how the machine-learning and rule-based approaches were used in combination; this is described in further technical detail in online supplemental file 3.

Two researchers (NC, AK) manually calculated precision performance for both versions of the application on 100 personal history documents (in domain testing data) and 100 other free-text document types (out domain test data) which had at least one occupation extraction and were previously unseen by the application in development. While both application versions performed well when text mining occupations from these test sets (precision ≥0.79, further detail in online supplemental file 3), the application with machine-learning approaches performed at the highest level of precision when assigning the occupation relation. The research team concluded from this testing phase that the application with combined

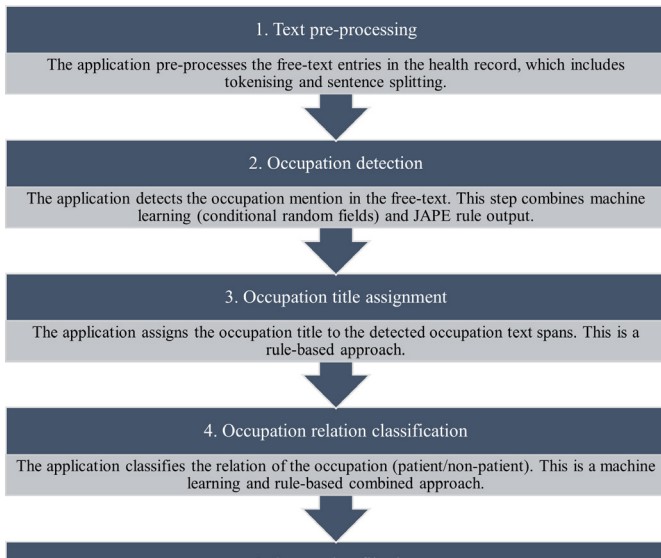

1. Text pre-processing

The application pre-processes the free-text entries in the health record, which includes tokenising and sentence splitting.

2. Occupation detection

The application detects the occupation mention in the free-text. This step combines machine learning (conditional random fields) and JAPE rule output.

3. Occupation title assignment

The application assigns the occupation title to the detected occupation text spans. This is a rule-based approach.

4. Occupation relation classification

The application classifies the relation of the occupation (patient/non-patient). This is a machine learning and rule-based combined approach.

5. Occupation filtering

The application filters out common false positives and health/social care occupations are not assigned to the patient, as part of a rule-based post-processing step.

**Figure 2** The process undertaken by the occupation application when text mining occupations from the clinical free-text field.

machine-learning and rule-based approaches was most appropriate, as this pipeline performed best at assigning the occupation relation.

### The healthcare occupation filter

The evaluation of the application performance over CRIS documents revealed that the most common false positives were extractions where the healthcare professional involved in the patient's care was incorrectly annotated as the patient's occupation (96% of annotations manually checked were health/social care occupations). To deal with this issue, health and social care occupations were added to a filter. The application then implemented a rules-based step where the filtered healthcare occupations were prevented from being annotated as belonging to the patient. Occupations added to this filter included variations on terms for psychiatrists and doctors, therapists, nurses and social workers, following the checking of 2390 documents to confirm that these were common false positives.

### Application implementation and testing

The final version of the text mining application with the healthcare filter applied was run over 10 free-text fields, including those where personal history sections were found, in the records of all patients on the CRIS case register aged 16 years and above. The fields included sections of the record such as discharge summaries, attachments, events and risk assessments (more detail in online supplemental file 1). The application was evaluated on a total of 866 documents: 666 gold-standard annotated personal history documents (test corpus 1), and 200 previously unannotated random personal history documents from the CRIS dataset at the time of

the application run (test corpus 2). Test corpus 1 was evaluated on GATE, and test corpus 2 was manually checked for occupations and then cross-referenced with the application output. The performance metrics considered the precision and recall level for the annotations made by the application, where both the occupation annotation and the relation classification needed to be accurate to be considered a 'true positive'. It was not feasible in this study to randomly select non-personal history documents for evaluation as patient occupations were rarely mentioned in the record compared with other information (eg, medication). As the application extracted an annotation entitled 'other', 200 of these annotations were manually checked for precision to further investigate these instances where the application was unable to assign an occupation title.

The EHR in the present study contains a structured field to record occupation: the 'Employment-ID'. This was explored on the CRIS platform using SQL queries. The proportion of completed 'Employment-IDs' from the records of all patients over the age of 16 years in January 2020 was extracted. The text mining application was simultaneously run over clinical records through CRIS, and the extracted patient occupations were converted into an SQL table. Sociodemographic, clinical and service contact data were also extracted from the structured fields of records using SQL queries. Data were exported to and analysed in STATA V.15 to examine predictors of occupational data extraction using logistic regression models. This included the patient's age at time of occupation extraction, gender, marital status, ethnicity, Index of Multiple Deprivation (IMD) score and primary diagnosis. Indicators of service contact included number of events in the record, number of face-to-face events in the record, number of spaces in the free-text fields of the record (as a proxy for word count), number of active days under SLaM services and number of inpatient bed-days. These variables were transformed into categories, for example, IMD scores were categorised into quartiles of local neighbourhood deprivation. Where data were missing for the extracted variables, this was coded as a 'not known' category for each variable.

Logistic regression models examined crude associations between the sociodemographic, clinical and service contact variables (predictors), and the recording of at least one patient occupation (outcome) from either the structured or free-text fields. The null hypothesis was that none of the predictors would be associated with occupation recording. First, models were adjusted for amount of contact the patient had with services. Fully adjusted models accounted for all other sociodemographic and clinical variables. Across all models, likelihood ratio tests were conducted to test the overall association between the variable and occupation recording. The aim of this analysis was to ascertain the characteristics of patients who had occupation recorded in their health record.

## Patient and public involvement

The proposal for this study was reviewed and approved by the patient-led CRIS oversight committee prior to the commencement of the project. No other consultations were made with patients or the public during the process of the study.

# RESULTS
## Annotating occupation

When double annotating 200 personal history documents, two annotators reached a Cohen's kappa agreement[24] of 0.77 for occupation title annotations and 0.72 for occupation relation annotations. Disagreements between annotators included instances where sentences posed unclear or vague references to occupation: for example, in the sentence, 'she did several things, such as cleaning, cooking', it was not clear whether these were domestic tasks or occupation descriptions, demonstrating the complexity of annotating occupation from text. Nonetheless, the Cohen's kappa agreement suggested that occupation could be annotated reasonably consistently across annotators using the annotation guidelines.

## Application development

The application reached a precision level of 0.88 and a recall level of 0.90 on the validation set of documents (N=77). The developed application process with combined rule-based and machine-learning approaches is described in figure 2.

## Application performance

When applied to the gold-standard annotated personal history documents (test corpus 1) on GATE, the application performed at a precision level of 0.79 and a recall level of 0.77. Two-hundred personal history documents were manually checked for occupations and then cross-referenced with the application output (test corpus 2): when considering patient occupations only, the application reached a precision level of 0.77 and recall level of 0.79. An extraction of 'other' as an occupational category was excluded from subsequent analysis, as the check of 200 annotations showed that this annotation only reached a precision level of 0.23 and often referenced job-seeking or non-work behaviours, for example, 'working on his anxiety'.

## Application implementation

Figure 3 shows the study population selection process for the implementation of the application over the CRIS case register, leading to an overall sample size of 341 720 patients.

## Descriptives

The demographics of the study population at the time of occupation extraction are described in table 1, as well as patient diagnostic categories and two indicators of the amount of service contact the patient has had: the number of 'events' entries added to the EHR and number

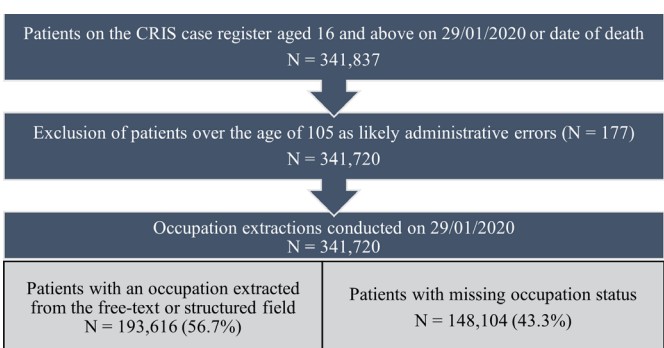

**Figure 3** The study population selection and extraction results from text mining occupations from the Clinical Record Interactive Search (CRIS) case register.

of inpatient bed-days. The three other extracted indicators for service contact (number of 'face-to-face events', total active days under SLaM mental health services and number of spaces in the text in the record) were excluded from analysis due to collinearity with the 'events' variable.

## Occupation extractions

The structured field for employment was populated for 46 705 (13.7%) patients. Prior to the implementation of the healthcare filter, 81.5% patients had at least one patient-occupation extraction. When using the final version of application to extract occupations from the free-text fields with the healthcare filter applied, this recalled at least one patient-related occupation for 184 521 patients (54.0%). By combining structured field and text mined occupations, patient-related occupations were retrieved for 193 616 patients (56.7%).

The structured field for occupation included 13 categories for occupational status, for example, 'unemployed' or 'paid employment'. In contrast, the text mining application retrieved 72 955 different patient-related occupation types. In total, there were 3 957 959 patient-related occupation extractions. Multiple occupation types were often extracted per patient (median=4, IQR=6).

The top five extracted occupations across the total sample of 341 720 patients were: student (n=98 719, 28.9%), unemployed (n=97 809, 28.6%), carer (n=61 893, 18.1%), self-employed (n=36 506, 10.8%) and retired (n=33 518, 9.8%). The less frequent extractions tended to be more specific occupation types, for example, 'retail worker' and 'banker'. The application also extracted undocumented ways of making money, including 'drug dealer' and 'sex worker'.

## Associations with occupation recording

Patients were split into binary groups: those who had an occupation recorded either in the structured field or free-text (n=193 616, 56.7%), and patients who did not have occupation recorded, that is, missing occupational data (n=148 104, 43.3%). Logistic regressions were used to examine sociodemographic, clinical and service contact associations with recorded occupations (table 2).

**Table 1** Sociodemographic and clinical features of the Clinical Record Interactive Search case register*

| | Number of patients (%) (total N=341 720) |
|---|---|
| **Age** | |
| 16–29 | 84 181 (24.63) |
| 30–49 | 123 216 (36.06) |
| 50–69 | 79 880 (23.38) |
| 70–89 | 43 852 (12.83) |
| 90+ | 10 591 (3.10) |
| **Gender** | |
| Male | 166 480 (48.72) |
| Female | 175 007 (51.21) |
| Other/not known | 233 (0.07) |
| **Ethnicity** | |
| White British | 136 289 (39.88) |
| Irish | 5182 (1.70) |
| Black Caribbean | 34 229 (10.02) |
| Black African | 15 654 (4.58) |
| Indian | 4345 (1.27) |
| Pakistani | 1852 (0.54) |
| Bangladeshi | 1088 (0.32) |
| Chinese | 1124 (0.33) |
| Other Asian | 5500 (1.61) |
| Other ethnic group | 19 650 (5.75) |
| Other white | 22 076 (6.46) |
| Mixed | 1879 (0.55) |
| Not known | 92 222 (26.99) |
| **Marital status** | |
| Married/civil partnership/cohabiting | 46 617 (13.64) |
| Divorced/separated/civil partnership dissolved | 17 309 (5.07) |
| Widowed | 15 758 (4.61) |
| Single | 141 111 (41.29) |
| Not known | 120 925 (35.39) |
| **Local quartiles of neighbourhood deprivation** | |
| Least deprived | 79 537 (23.28) |
| 3rd quartile | 80 049 (23.43) |
| 2nd quartile | 79 767 (23.34) |
| Most deprived | 79 829 (23.36) |
| Address not known | 22 538 (6.60) |
| **Primary diagnosis** | |
| F30–F39: mood (affective) disorders | 37 796 (11.06) |
| F00–F09: organic, including symptomatic, mental disorders | 29 801 (8.72) |
| F10–F19: mental and behavioural disorders due to psychoactive substance misuse | 27 870 (8.16) |

Continued

**Table 1** Continued

| | Number of patients (%) (total N=341 720) |
|---|---|
| F20–F29: schizophrenia, schizotypal and delusional disorders | 18 253 (5.34) |
| F40–F49: neurotic, stress-related and somatoform disorders | 31 962 (9.35) |
| F50–F59: behavioural syndromes associated with physiological disturbances and physical factors | 9166 (2.68) |
| F60–F69: disorders of adult personality and behaviour | 6605 (1.93) |
| F70–F79: mental retardation | 2732 (0.80) |
| F80–F89: disorders of psychological development | 5874 (1.72) |
| F90–F98: behavioural and emotional disorders with onset usually occurring in childhood and adolescence | 12 028 (3.52) |
| Other diagnosis | 83 847 (24.54) |
| Not known | 75 786 (22.18) |
| **Quartiles of 'events' entered into the health record** | |
| No events | 50 673 (14.83) |
| Least events (1–3) | 86 818 (25.41) |
| 2nd quartile (4–10) | 62 804 (18.38) |
| 3rd quartile (11–40) | 68 774 (20.13) |
| Most events (41+) | 72 651 (21.26) |
| **Inpatient bed-days** | |
| No inpatient admissions | 311 099 (91.04) |
| Low (1–2) | 1937 (0.50) |
| Moderate (3–31) | 10 587 (3,10) |
| High (32+) | 18 337 (5.37) |

*At the time of the occupation application run (29 January 2020).

Across all models, all predictors were strongly associated with a recording of occupation even after fully adjusting for all other variables (likelihood ratio tests p<0.0001). When key sociodemographic data were missing from the record, the odds of occupational data being recorded decreased: for example, where the marital status of the patient was 'not known', the fully adjusted OR for a recording of an occupation was 0.49 (95% CI 0.47 to 0.50) compared with patients who were recorded as married/in a civil partnership/cohabiting. Female patients were significantly less likely to have occupation recorded compared with male patients, and older patients were most likely to have occupational data recorded compared with the youngest patients. Compared with patients of white British ethnicity, patients of Irish, black Caribbean or black African ethnicity were more likely to have an occupation recorded; while Indian, Pakistani, Chinese,

**Table 2**  Results from crude and multivariable logistic regression analyses examining predictors of occupation recording from the Clinical Record Interactive Search case register*

| | N (%) with at least one occupation retrieved by structured field/text mining extractions | OR (95% CI) | aOR† (95% CI) | aOR‡ (95% CI) |
|---|---|---|---|---|
| **Age** | | | | |
| 16–29 | 41 653 (49.48) | Reference | Reference | Reference |
| 30–49 | 68 422 (55.53) | 1.27 (1.25 to 1.30) | 1.56 (1.53 to 1.59) | 1.72 (1.68 to 1.75) |
| 50–69 | 49 289 (61.70) | 1.65 (1.61 to 1.68) | 1.98 (1.93 to 2.02) | 2.19 (2.14 to 2.25) |
| 70–89 | 27 175 (61.97) | 1.66 (1.63 to 1.70) | 1.71 (1.67 to 1.76) | 1.60 (1.54 to 1.65) |
| 90+ | 7077 (66.82) | 2.06 (1.97 to 2.15) | 2.14 (2.04 to 2.24) | 2.00 (1.89 to 2.11) |
| **Gender** | | | | |
| Male | 96 141 (57.75) | Reference | Reference | Reference |
| Female | 97 443 (55.68) | 0.92 (0.91 to 0.93) | 0.88 (0.87 to 0.90) | 0.87 (0.85 to 0.88) |
| Other/not known | 32 (13.73) | 0.12 (0.08 to 0.17) | 0.10 (0.07 to 0.15) | 0.16 (0.10 to 0.24) |
| **Ethnicity** | | | | |
| White British | 91 575 (67.19) | Reference | Reference | Reference |
| Irish | 4303 (74.04) | 1.39 (1.31 to 1.48) | 1.24 (1.17 to 1.33) | 1.23 (1.15 to 1.31) |
| Black Caribbean | 24 753 (72.32) | 1.28 (1.24 to 1.31) | 0.99 (0.96 to 1.02) | 1.06 (1.03 to 1.09) |
| Black African | 11 341 (72.45) | 1.28 (1.24 to 1.33) | 1.07 (1.03 to 1.11) | 1.12 (1.07 to 1.17) |
| Indian | 2876 (66.19) | 0.96 (0.90 to 1.02) | 0.91 (0.85 to 0.97) | 0.91 (0.85 to 0.98) |
| Pakistani | 1185 (63.98) | 0.87 (0.79 to 0.95) | 0.81 (0.73 to 0.90) | 0.82 (0.74 to 0.91) |
| Bangladeshi | 719 (66.08) | 0.95 (0.84 to 1.08) | 0.90 (0.78 to 1.03) | 0.94 (0.82 to 1.08) |
| Chinese | 690 (61.39) | 0.78 (0.69 to 0.88) | 0.73 (0.65 to 0.84) | 0.81 (0.71 to 0.92) |
| Other Asian | 3543 (64.42) | 0.88 (0.84 to 0.94) | 0.82 (0.78 to 0.87) | 0.85 (0.80 to 0.91) |
| Other ethnic group | 11 768 (59.89) | 0.73 (0.71 to 0.75) | 0.77 (0.75 to 0.80) | 0.75 (0.72 to 0.77) |
| Other white | 14 610 (66.18) | 0.96 (0.93 to 0.98) | 0.94 (0.91 to 0.97) | 0.97 (0.94 to 1.00) |
| Mixed race | 1197 (63.70) | 0.86 (0.78 to 0.94) | 0.68 (0.61 to 0.75) | 0.78 (0.70 to 0.87) |
| Not known | 25 056 (27.17) | 0.18 (0.18 to 0.19) | 0.31 (0.31 to 0.32) | 0.50 (0.49 to 0.51) |
| **Marital status** | | | | |
| Married/civil partnership/cohabiting | 31 037 (66.58) | Reference | Reference | Reference |
| Divorced/separated/civil partnership dissolved | 13 346 (77.10) | 1.69 (1.62 to 1.76) | 1.47 (1.40 to 1.53) | 1.41 (1.35 to 1.47) |
| Widowed | 11 309 (71.77) | 1.28 (1.23 to 1.33) | 1.05 (1.00 to 1.09) | 1.05 (1.01 to 1.10) |
| Single | 98 841 (70.04) | 1.17 (1.15 to 1.20) | 1.02 (1.00 to 1.05) | 1.24 (1.21 to 1.27) |
| Not known | 39 083 (32.32) | 0.24 (0.23 to 0.25) | 0.33 (0.32 to 0.33) | 0.49 (0.47 to 0.50) |
| **Local quartiles of neighbourhood deprivation** | | | | |
| Least deprived | 48 155 (60.54) | | Reference | Reference |
| 3rd quartile | 47 583 (59.44) | 0.96 (0.94 to 0.97) | 0.97 (0.95 to 0.99) | 0.96 (0.94 to 0.99) |
| 2nd quartile | 45 842 (57.47) | 0.88 (0.86 to 0.90) | 0.94 (0.91 to 0.96) | 0.93 (0.91 to 0.95) |
| Most deprived | 41 800 (52.36) | 0.72 (0.70 to 0.73) | 0.89 (0.87 to 0.91) | 0.88 (0.86 to 0.90) |
| Address not known | 10 236 (45.42) | 0.54 (0.53 to 0.56) | 0.70 (0.67 to 0.72) | 0.77 (0.74 to 0.80) |
| **Diagnosis** | | | | |
| F30–F39: mood (affective) disorders | 27 057 (71.59) | Reference | Reference | Reference |
| F00–F09: organic, including symptomatic, mental disorders | 20 269 (68.01) | 0.84 (0.82 to 0.87) | 0.91 (0.88 to 0.94) | 0.71 (0.68 to 0.74) |

**Table 2** Continued

| | N (%) with at least one occupation retrieved by structured field/text mining extractions | OR (95% CI) | aOR† (95% CI) | aOR‡ (95% CI) |
|---|---|---|---|---|
| F10–F19: mental and behavioural disorders due to psychoactive substance misuse | 18 150 (65.12) | 0.74 (0.72 to 0.77) | 0.71 (0.68 to 0.73) | 0.47 (0.45 to 0.49) |
| F20–F29: schizophrenia, schizotypal and delusional disorders | 14 645 (80.23) | 1.61 (1.54 to 1.68) | 0.87 (0.83 to 0.91) | 0.78 (0.74 to 0.82) |
| F40–F49: neurotic, stress-related and somatoform disorders | 19 920 (62.32) | 0.66 (0.64 to 0.68) | 0.75 (0.72 to 0.77) | 0.76 (0.73 to 0.79) |
| F50–F59: behavioural syndromes associated with physiological disturbances and physical factors | 5287 (57.68) | 0.54 (0.52 to 0.57) | 0.65 (0.62 to 0.68) | 0.68 (0.64 to 0.72) |
| F60–F69: disorders of adult personality and behaviour | 4739 (71.75) | 1.01 (0.95 to 1.07) | 0.68 (0.64 to 0.73) | 0.77 (0.72 to 0.82) |
| F70–F79: mental retardation | 2277 (83.35) | 1.99 (1.79 to 2.20) | 1.81 (1.63 to 2.03) | 1.69 (1.51 to 1.90) |
| F80–F89: disorders of psychological development | 4377 (74.78) | 1.16 (1.09 to 1.24) | 1.22 (1.14 to 1.30) | 1.78 (1.66 to 1.92) |
| F90–F98: behavioural and emotional disorders with onset usually occurring in childhood and adolescence | 8754 (72.78) | 1.06 (1.01 to 1.11) | 1.25 (1.19 to 1.32) | 1.84 (1.74 to 1.93) |
| Other diagnosis | 43 787 (52.22) | 0.43 (0.42 to 0.45) | (0.68 to 0.72) | 0.76 (0.73 to 0.78) |
| Not known | 24 354 (32.14) | 0.19 (0.18 to 0.19) | 0.44 (0.43 to 0.45) | 0.66 (0.64 to 0.68) |
| Quartiles of 'events' entered into the health record | | | | |
| No events | 12 012 (23.70) | Reference | Reference | Reference |
| Least events | 35 009 (40.32) | 2.17 (2.12 to 2.23) | 2.18 (2.13 to 2.23) | 1.75 (1.70 to 1.79) |
| 2nd quartile | 34 368 (54.72) | 3.89 (3.79 to 3.99) | 3.89 (3.79 to 3.99) | 2.79 (2.71 to 2.87) |
| 3rd quartile | 49 237 (71.59) | 8.11 (7.90 to 8.33) | 8.06 (7.85 to 8.28) | 5.01 (4.86 to 5.16) |
| Most events | 62 990 (86.70) | 20.98 (20.37 to 21.60) | 18.89 (18.29 to 19.50) | 9.77 (9.43 to 10.1) |
| Inpatient bed-days | | | | |
| No inpatient admissions | 167 213 (53.75) | Reference | Reference | Reference |
| Low (1–2) | 1408 (82.97) | 4.19 (3.69 to 4.76) | 1.87 (1.64 to 2.14) | 1.68 (1.47 to 1.93) |
| Moderate (3–31) | 8714 (82.31) | 4 (3.81 to 4.21) | 1.06 (1.00 to 1.11) | 1.01 (0.95 to 1.07) |
| High (32+) | 16 281 (88.79) | 6.81 (6.51 to 7.14) | 1.57 (1.49 to 1.66) | 1.32 (1.25 to 1.39) |

*All variables listed in this table had a strong association with the outcome variable (p<0.0001), assessed by likelihood ratio tests.
†Adjusted for service contact variables (number of events and inpatient bed-days).
‡Adjusted for all other variables in the table.
aOR, adjusted OR.

mixed race or patients recorded as being from 'other' Asian or ethnic groups were less likely to have occupation recorded. The odds of having occupation recorded were significantly lower for patients who were living in the most deprived local areas compared with the most affluent areas. Generally, patients with a primary diagnosis of an affective disorder had a higher odds of an occupation extraction than patients with other diagnoses, including organic disorders. In the crude logistic regression models, patients diagnosed with schizophrenia, schizotypal or delusional disorders were more likely to have occupation

extracted (OR 1.61, 95% CI 1.54 to 1.68). However, once adjusting for amount of contact with services, these patients were significantly less likely to have occupation extracted compared with patients with affective disorders (adjusted OR 0.87, 95% CI 0.83 to 0.91).

## DISCUSSION

Annotating and extracting occupation from the free-text fields in clinical records are challenging tasks. We have developed a tool to text mine patient occupations

with a good degree of confidence from a mental health EHR, and applied this at scale over a large EHR in south London. An important finding was that we could retrieve over double the number of patient occupations using text mining methodology than when using pre-existing structured fields alone. We could also access a much wider diversity of occupation types: this further detail on occupations held by patients opens up the possibility for the translation of occupations onto social class schema, which would not have been possible with the limited structured field categories. The most prevalent patient occupations were 'student' and 'unemployed'. There were differences between patients who had occupation recorded and patients whose occupation data remained missing: patients with occupations recorded were more likely to be of an older age, male, divorced/separated, living in areas of lower deprivation and have more contact with mental health services. Across ethnic minority groups, there were mixed findings relating to the recording of occupation. Compared with white British patients, Irish, black Caribbean and black African patients were slightly more likely to have a recording of occupation, whereas all other ethnic minority groups were less likely to have a recording. Although it is possible that some of the demographic associations with the recording of occupation in the case notes were impacted by residual confounding in adjusted models, these findings may also indicate disparities relating to how occupations are assessed and recorded in the clinical record and should be explored in future work, particularly given the strong correlation of employment with recovery within the context of mental disorders.

This study broadly supports the work of other studies which indicate that clinicians mostly describe occupation in the free-text of EHR systems, when these are available, rather than structured fields.[10] This study is the first of its kind to text mine patient occupations from a mental healthcare EHRs. There have been several previous efforts to extract patient occupations from other healthcare free-text notes. Occupations have been text mined from general medical clinical text; however, in these studies the algorithms reached low levels of performance, largely due to a lack of training data.[25 26] Dehghan and colleagues text mined occupation from the clinical records of patients with cancer in the UK, reaching similar precision and recall levels to the present study.[27] However, none of these applications distinguished between text mining occupations belonging to the patient and other relations, had the scope of applying and testing the text mining methodology at scale across the EHR or examined associations with extracted versus missing occupational data. The present application therefore represents significant progress in our ability to text mine patient occupations from the EHR and furthers our understanding of what this may mean in practice.

We found that text mining greatly increased our retrieval of patient occupations in this psychiatry EHR database. Psychiatric notes may be more detailed than other types of healthcare text (for example, in general medicine) when describing the patient's occupation, as this often forms part of psychiatric history taking and assessment. We found that a sizeable proportion of patients over CRIS have at some point been a student or unemployed. A separate NLP application being developed using CRIS data (by author JS) will be able to interrogate this student group further by extracting the patient's level of educational attainment, which will complement the present application. There is also scope to explore older groups of patients who are students but are also working using this methodology. Our finding that unemployment was a dominant occupational category is consistent with previous research, in that unemployment levels are elevated particularly for those with severe mental illnesses compared with the general population.[1 2] It may also be the case that some patients in this group are formally unemployed but are working in more informal, undocumented ways to make money. This application identified some informal occupations, which provides interesting avenues for further research.

One limitation of our approach is that we could not distinguish the temporality of occupations—whether they were currently or previously held by the patient. While developing the annotation guidelines, we found that the text was unlikely to be sufficient to assess temporality, as it was often not explicitly stated when the patient started or left an occupation, or how long they have held a position for. Multiple occupations were often extracted for a single patient, adding to the complexity. While there is work ongoing to use NLP to detect temporality in psychiatric healthcare text,[28] this remains a challenge and is a potential avenue for further work that is beyond the scope of this study. As this application was developed at a single site in the UK, the generalisability of the application may be reduced, first to text in the English language and second to this catchment area. As it was not possible to assign health and social care occupations to patients with reasonable confidence, we will also be missing patients who hold these occupations; however, we are planning further work to develop this aspect of the application. Notwithstanding these limitations, this application was developed through an extensive process of training and testing using a large corpus leading to the application of text mining algorithms for occupation at scale. This methodology is already revealing the kinds of occupations held by patients using secondary mental health services.

The development of this application has numerous implications. First, this application will be valuable in allowing researchers to examine relationships between occupation and health in large psychiatric case registers. For example, work is currently underway using this application to investigate predictors of unemployment in a cohort of patients with severe mental illness.[29] As CRIS-like systems are in use over several sites in the UK, there is the scope to test and implement this application in other mental healthcare providers using similar EHR platforms. This application could also have potential

practical implications including identifying unemployed patients to target interventions such as Individual Placement and Support and retrieving occupational distributions for audits and organisational monitoring in NHS mental health Trusts. Lastly, this application may have implications beyond mental health research and text, notably in research on industrial injuries, although this requires further testing.

There is room for further progress in this application as the NLP field further develops, including identifying the temporality of occupations and improving relation classification for health and social care occupations. We plan to develop methodology to ascertain the occupational social class of patients, using the large diversity of occupations extracted, to further inform health inequalities research specific to mental health. Future studies implementing this application in other CRIS systems may be able to investigate the transferability of the application to other NHS sites in the UK that serve different patient populations. Overall, we hope that this approach will prove useful in addressing our understanding of the interactions between occupation and health in those with mental illness.

**Author affiliations**
[1]Institute of Psychiatry, Psychology and Neuroscience, King's College London, London, UK
[2]Department of Computer Science, University of Sheffield, Sheffield, UK
[3]South London and Maudsley NHS Foundation Trust, London, UK
[4]King's College Hospital NHS Trust, London, UK
[5]Economic and Social Research Council (ESRC) Centre for Society and Mental Health, King's College London, London, UK

**Acknowledgements** We appreciated the technical support from informatics personnel in the NIHR Maudsley Biomedical Research Centre and the University of Sheffield. We would also like to thank Shirlee MacCrimmon for her assistance with annotations during the annotation guideline development process.

**Contributors** The study was conceived by JD-M, AK, RS, AR, SH, BG and LH-A. Personal history sections of documents were extracted using an application developed by DC. Manual annotations to develop the annotation guidelines and produce the test and training data were conducted by AK, AR, ET, ZC, KB and SM (acknowledgements). The application was developed by XS, with feedback from AK, NC, JD-M, RS, AR and SH. The application was implemented over the EHR by DC and JS. The application was evaluated by AK and NC. The missing data analysis was conducted by NC and JD-M. The paper draft was led by NC, JD-M and AK; and was critically reviewed and edited by all authors (AK, XS, AR, ET, RS, ZC, KB, DC, JS, BG, LH-A, SH).

**Funding** This work was supported by the Clinical Record Interactive Search system which is funded by the National Institute for Health Research (NIHR) Biomedical Research Centre at South London and Maudsley National Health Service (NHS) Foundation Trust and King's College London and a joint infrastructure grant from Guy's and St Thomas' Charity and the Maudsley Charity. Specific application development was supported by the Health Foundation, in a grant held by JD-M. NC is funded by the Economic and Social Research Council for a PhD studentship under the London Interdisciplinary Social Science Doctoral Training Partnership (ES/P000703/1). JD-M is funded by the Health Foundation working together with the Academy of Medical Sciences, for a Clinician Scientist Fellowship and by the ESRC in relation to the SEP-MD Study (ES/S002715/1) and part supported by the ESRC Centre for Society and Mental Health at King's College London (ESRC Reference: ES/S012567/1). RS is part-funded by: (1) the NIHR Biomedical Research Centre at the South London and Maudsley NHS Foundation Trust and King's College London; (2) a Medical Research Council (MRC) Mental Health Data Pathfinder Award to King's College London; (3) an NIHR Senior Investigator Award; (4) the NIHR Applied Research Collaboration south London (NIHR ARC south London) at King's College Hospital NHS Foundation Trust. SH is part-funded by the NIHR Biomedical Research Centre at the South London and Maudsley NHS Foundation Trust and King's College London; ESRC Centre for Society and Mental Health at King's College London (ESRC Reference: ES/S012567/1); and the Wellcome Trust (203380Z16Z).

**Disclaimer** The funders had no role in the study design, data collection and analysis, decision to publish or preparation of the manuscript. The views expressed are those of the author(s) and not necessarily those of the NHS, the NIHR, the MRC, the Department of Health, the ESRC or King's College London.

**Competing interests** None declared.

**Patient consent for publication** Not required.

**Ethics approval** The SLaM Case Register has been approved as a source of de-identified data for secondary analyses (Oxford Research Ethics Committee C, reference 18/SC/0372).

**Provenance and peer review** Not commissioned; externally peer reviewed.

**Data availability statement** Data are available upon reasonable request. We are unable to place test data in the public domain because these comprise patient information, but document IDs used in application development and testing have been archived and researchers may apply for approval to access these or other CRIS data. This application is also being put into production for researchers to use in the Biomedical Research Centre. More information and contact details can be found at brc.slam.nhs.uk/about/core-facilities/cris.

**ORCID iDs**
Natasha Chilman http://orcid.org/0000-0002-9661-5098
Xingyi Song http://orcid.org/0000-0002-4188-6974
Angus Roberts http://orcid.org/0000-0002-4570-9801
Esther Tolani http://orcid.org/0000-0002-7415-0859
Robert Stewart http://orcid.org/0000-0002-4435-6397
Zoe Chui http://orcid.org/0000-0001-6844-6779
Karen Birnie http://orcid.org/0000-0003-4123-1676
Lisa Harber-Aschan http://orcid.org/0000-0002-6464-485
Billy Gazard http://orcid.org/0000-0002-7562-539
David Chandran http://orcid.org/0000-0002-0123-666X
Stephani Hatch http://orcid.org/0000-0001-9103-2427
Anna Kolliakou http://orcid.org/0000-0003-1234-4129
Jayati Das-Munshi http://orcid.org/0000-0002-3913-6859

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
