## [Reviewer comments · BMJ Open]

ARTICLE DETAILS

TITLE (PROVISIONAL)	Text-mining occupations from the mental health electronic health record: a natural language processing approach using records from the Clinical Record Interactive Search (CRIS) platform in south London, UK.
AUTHORS	Chilman, Natasha; Song, Xingyi; Roberts, Angus; Tolani, Esther; Stewart, Robert; Chui, Zoe; Birnie, Karen; Harber-Aschan, Lisa; Gazard, Billy; Chandran, David; Sanyal, Jyoti; HATCH, STEPHANI; Kolliakou, Anna; Das-Munshi, Jayati

VERSION 1 – REVIEW

REVIEWER	A Abbe Institut national de la santé et de la recherche médicale, France
REVIEW RETURNED	27-Jul-2020

GENERAL COMMENTS	Thank you for giving me the opportunity to review “Text-mining occupations from the mental health electronic health record: a natural language processing approach using records from the Clinical Record Interactive Search (CRIS) platform in South London, UK.”. I am very pleased to read this paper due to the high quality of evaluation methods used. In addition, it is well written. I have very few comments. Firstly, the number of text used to evaluate the quality of detection is low. It would be necessary to increase the number of document screened to ensure precision and recall levels. Secondly, I am a little bit frustrated to see that GATE was used. It is a Java platform not widely used to perform NLP. I would have preferred R software that is less specific to the domain but easier to extract and model text. In addition, authors did not explain how GATE structured the text (using lemmatisation, stemming, ...). I would recommend to add a more information about it in methods and maybe limitation sections.
---

REVIEWER	Mike Conway University of Utah, United States
REVIEW RETURNED	16-Aug-2020

GENERAL COMMENTS	# [REVIEW] Text-mining occupations from the mental health electronic health record: a natural language processing approach using records from the Clinical Record Interactive Search (CRIS) platform in South London, UK. BMJ Open - Aug 2020
---

	## SUMMARY The goal of this work is the automatic identification of occupational information — both job titles (e.g. “builder”) and broad areas of work (e.g. construction) — from clinical notes derived from South London & Maudsley NHS Trust psychiatric notes. The motivation of the study is that occupational information is likely to be salient in studies focussed on the social determinants of health, particularly mental health. The research used a combination of CRF & rule-based algorithms to identify occupations. When structured data was used, 14% of patients had occupation recorded. This proportion increased to 57% using the NLP algorithm. The algorithm is reliable enough to be useful (i.e. 0.78 F-score), though there are issues that may limit portability to different clinical contexts. This is a valuable research endeavour focussed on a currently understudied task. The resulting, validated tool could potentially be useful for a number of different tasks (e.g. generating occupational cohorts) that are currently quite difficult to conduct. ## MAJOR COMMENTS  1. The ML component is currently underspecified (e.g. what was the implementation used?). For example, CRFs are mentioned only once (page 8 of 54). 2. The NLP literature review could be strengthened significantly 3. The manuscript — although generally easy to understand — does have numerous grammar issues. ## MINOR COMMENTS  4. Limitations of the work (e.g. the difficulties involved in establishing whether a patient holds a health/social care occupation) are clearly outlined 5. Abstract -> Outcomes. This section seems somewhat schematic 6. p6 of 54. I'd suggest extending the NLP lit review beyond just mental health related NLP. The work described in the paper could be used for a bunch of different research questions (e.g. industrial injury analysis) that are not related to mental health. 7. p8 of 54. paragraph 1, suggest including reference to kappa statistic here 8. p10 of 54. “Models were firstly adjusted for...” grammar 9. p14 of 54 “a cutting edge methodology” - I'm not sure that the work can be reasonably described as cutting-edge from a methodological perspective, though it is innovative in terms of the application area 10. Are you planning on making the trained models/rule set available to the community? This would increase the impact of the work substantially.
--	---

REVIEWER	Hong-Jie Dai National Kaohsiung University of Science and Technology, Taiwan
REVIEW RETURNED	17-Aug-2020

GENERAL COMMENTS	The authors presented a study to develop and evaluate a natural language processing (NLP) system based on GATE software to
--

	extract occupation information from clinical records. They then compared the results from unstructured text to the structured fields and observed that the text-mined results can complement that from structured fields by 43%. Overall the paper is well written but I have the following concerns. First of all, it is unclear for me how the authors used the annotated data to develop their NLP systems. It seems like that they generated several versions of the corpora for developing their GATE-based pipeline. At first, they have 600 documents followed by 1000 documents. But they only used 77 documents as the training set for the development of the versions of their NLP systems; one is machine learning approaches and another is the rule-based approach. It is unusual to use such a small amount of data as the training set considering that they have a total of 1600 documents. The performance reported on page 8 and supplementary file (SF) 3 is neither not reliable because the performance shown in Table 1 of SF3 was evaluated on the 77 training set instead of an individual test set. The version number shown in Table 2 of SF3 is also meaningless for readers. From the descriptions, I suggested the authors to consider use more training set (maybe 1000 documents) to develop their system and evaluate its performance on the remaining documents (in this case 600). It is also unclear for me how they come to the conclusion that the combined approach was more appropriate for the occupation extraction task. The way to combine the two approaches is also unclear. The most valuable part of this work should be the association analysis between the extracted occupation and several pre-defined predictors. However, several NLP design trade-offs were made during the entire development lead to a doubt of the soundness of their conclusion. One significant limitation is the lack of the determination of temporal attribute of the extracted occupations. I think the authors should try to consider to solve it at first. One thing the authors can do maybe is to compare the results extracted by NLP system with that recorded in the structured fields. They could calculate the overlapping rate between the two sources and conduct some analysis to see how much extent NLP can support. Some minor comments Page 7 line 46. What is the recall rate of the developed NLP application for the extraction of the PH sections? Is there any citation for the application? Please double-check the abbreviations used in the draft. I cannot find the full names of some terms like SLAM.
--	--

REVIEWER	J. Zwaveling Leiden University Medical Center, Leiden, the Netherlands
REVIEW RETURNED	20-Aug-2020

GENERAL COMMENTS	I have no comments, since the manuscript is written very clearly and of outstanding quality. 1 minor remark: reference 10 is mentioned before reference 6 and 7 in the introduction.
---

VERSION 1 – AUTHOR RESPONSE

Reviewer: 1

Reviewer Name: A Abbe

“I am very pleased to read this paper due to the high quality of evaluation methods used. In addition, it is well written. I have very few comments.

Firstly, the number of text used to evaluate the quality of detection is low. It would be necessary to increase the number of document screened to ensure precision and recall levels.”

Following this feedback, we now explicitly state that we evaluated the application on “a total of 866 documents” (page 8, paragraph 4). In the original manuscript, we described the total number of documents in each of the two test sets separately, rather than this total. It was not feasible in this project to annotate more documents to a gold-standard due to resources, as annotating is time-consuming and expensive to conduct. As the reviewer states that our evaluation methods were high quality, we hope that this further clarification that 866 documents were used in total demonstrates that a sufficient number of documents were used for the evaluation of the application.

“Secondly, I am a little bit frustrated to see that GATE was used. It is a Java platform not widely used to perform NLP. I would have preferred R software that is less specific to the domain but easier to extract and model text. In addition, authors did not explain how GATE structured the text (using lemmatisation, stemming, ...). I would recommend to add a more information about it in methods and maybe limitation sections.”

As the reviewer suggests, we have added more detail about the GATE framework’s capabilities in the methods section to justify why we have used this framework (page 7, paragraph 2). However, we respectfully disagree that GATE is not widely used to perform NLP. As we now also describe in the paper (page 7, paragraph 2), GATE is “a widely used NLP network with over 40 thousand downloads per version and a history of use in the UK national health services, amongst other sectors”. It has also been used in other NLP applications using clinical text from this specific NHS electronic health record system (references 12-16). A fairly recent systematic review of NLP systems in the clinical domain found a number of systems which used GATE frameworks (reference 17). The results from this review demonstrate that GATE is widely used to perform NLP. We agree with the reviewer in that R is less specific to the NLP domain, therefore GATE was chosen as the most appropriate platform on which to develop this application. Following the reviewer’s suggestion, we have provided more information on how GATE structured the text in Figure 2 in the paper (step 1: text pre-processing), and in a new detailed flow diagram of the GATE and NLP processes in supplementary file 3. We also provide a text description of how GATE structured the text in supplementary file 3.

Reviewer: 2

Reviewer Name: Mike Conway

“This is a valuable research endeavour focussed on a currently understudied task. The resulting, validated tool could potentially be useful for a number of different tasks (e.g. generating occupational cohorts) that are currently quite difficult to conduct.

The ML component is currently underspecified (e.g. what was the implementation used?). For example, CRFs are mentioned only once (page 8 of 54).”

We have now added to the machine-learning description of the application on page 8, paragraph 1 to specify the ML component of the application: the conditional random field classifier was used “to

identify occupation mentions in the text, and a support-vector machine-based classifier [was used] to identify the occupation relation.” As we wish to keep the paper accessible to all (including readers who are not trained in NLP methods), we have added a significant amount of technical detail to supplementary file 3 on the NLP pipeline.

“The NLP literature review could be strengthened significantly “ & “. I’d suggest extending the NLP lit review beyond just mental health related NLP. The work described in the paper could be used for a bunch of different research questions (e.g. industrial injury analysis) that are not related to mental health.”

We have developed the NLP literature review in the introduction (page 5, paragraph 1) to include previous research where NLP is applied in industrial/occupational research contexts. We have also added a sentence in the discussion to highlight that that this application could be utilised beyond mental health-related text, but this would require testing (page 14-15, paragraph 3).

“The manuscript — although generally easy to understand — does have numerous grammar issues.” p10 of 54. “Models were firstly adjusted for...” grammar”

We have corrected grammatical errors in the original manuscript (sentence structure on page 9, paragraph 3 as suggested by reviewer; in title South London changed to south London; sentence structure on page 5, paragraph 2; sentence structure page 9, paragraph 3).

“p8 of 54. paragraph 1, suggest including reference to kappa statistic here.”

Health/social care occupations were not double-annotated; therefore we do not have a kappa statistic of agreement for this phase of development. We report the kappa statistic for the double-annotations of personal history text.

“p14 of 54 “a cutting edge methodology” - I’m not sure that the work can be reasonably described as cutting-edge from a methodological perspective, though it is innovative in terms of the application area.”

We have changed this to ‘innovative’ as suggested (page 13, paragraph 1).

“Are you planning on making the trained models/rule set available to the community?”

We hope to make it publicly hosted in future, subject to permissions due to the sensitive nature of the data used for training and testing.

Reviewer: 3

Reviewer Name: Hong-Jie Dai

“It is unclear for me how the authors used the annotated data to develop their NLP systems. It seems like that they generated several versions of the corpora for developing their GATE-based pipeline. At first, they have 600 documents followed by 1000 documents. But they only used 77 documents as the training set for the development of the versions of their NLP systems; one is machine learning approaches and another is the rule-based approach. It is unusual to use such a small amount of data as the training set considering that they have a total of 1600 documents.”

We have developed the manuscript to provide more clarity on how the annotated data was used to train and test our application (page 7, paragraphs 1 and 2). The 600 documents which were annotated in the early stages of this project were not included in the gold-standard annotated corpus,

as these were annotated for the purpose of practice to develop the guidelines. As a consequence, these annotations do not follow the finalised version of the guidelines and are therefore not completed to a 'gold-standard'. A sentence has been added in the methods (page 7, paragraph 1) to clarify that the 600 documents were annotated to "practice annotating occupation from text and develop the annotation guidelines", and that the 1000 documents were annotated "to a gold standard". Following this reviewer comment, we corrected our manuscript as we originally reported that the 77 documents were used as a training set. After discussions within the research team, it has been clarified that these 77 documents were used as a validation set, not a training set. We used 257 of the gold-standard annotated documents as a training set. We now report that the training set comprised of 257 documents, and the validation set comprised of 77 documents (page 7, paragraph 2). We have also added this training set to Figure 1, step 4. We reserved the remaining gold-standard annotated corpus for application evaluation (n=666). Therefore, we now clearly report that all the 1000 annotated gold-standard documents were utilised to their full capacity in the development, training and testing of the application.

"The performance reported on page 8 and supplementary file (SF) 3 is neither not reliable because the performance shown in Table 1 of SF3 was evaluated on the 77 training set instead of an individual test set."

We have removed the application performance report on the 77-document validation set in supplementary file 3, and retained the performance on test data as the reviewer suggests. The performance reported on page 8 is from testing sets, not training sets: this has been clarified on page 8, paragraph 1 ("from these test sets").

"The version number shown in Table 2 of SF3 is also meaningless for readers."

We have removed the version number in this table.

"From the descriptions, I suggested the authors to consider use more training set (maybe 1000 documents) to develop their system and evaluate its performance on the remaining documents (in this case 600)."

As above, the 600 documents annotated at the beginning of application development were not annotated to a gold-standard. We used a total of 1000 annotated documents to develop and test our system, supplemented with 200 unannotated test documents (test set 2) which were cross-checked with the application output. In total, we use a total of 1,200 documents in application training, validation and testing.

"It is also unclear for me how they come to the conclusion that the combined approach was more appropriate for the occupation extraction task. The way to combine the two approaches is also unclear."

We continued with a combined approach because this approach performed best at classifying the occupation relation (this has been expanded on in the manuscript following this reviewer feedback on page 8, paragraph 2: "as this pipeline performed best at assigning the occupation relation"). Figure 2 describes which elements of the application annotation process are machine learning and rule-based and how these were combined (e.g. detecting occupation is a CRF and JAPE rule-based combined approach). We have now added a further description and pipeline diagram to supplementary file 3 to provide a more technical description of the application pipeline and how ML and rule-based approaches were combined.

"One significant limitation is the lack of the determination of temporal attribute of the extracted

occupations. I think the authors should try to consider to solve it at first. One thing the authors can do maybe is to compare the results extracted by NLP system with that recorded in the structured fields. They could calculate the overlapping rate between the two sources and conduct some analysis to see how much extent NLP can support.”

We agree that not being able to identify the temporality of occupations is a limitation of the current application, as described in the discussion. Unfortunately, to identify the temporality of occupations recorded within this specific application is beyond the scope of this paper and we have some doubts regarding the feasibility of this with the text available: when manually annotating and checking the text extractions throughout application development and evaluation (particularly during the annotation guideline development stage), it was noted that the text often did not describe temporality. For example, a clinician will report that the patient has been in a certain occupation, but will often not state how long they have been in that job or when it started/ended. Therefore, it is likely that the text is currently not sufficient in quality to identify occupation temporality. We have added to the methods and discussion to ensure that these limitations regarding temporality are more clearly highlighted (page 6, paragraph 1; page 14, paragraph 2).

We have considered the reviewer’s suggestion to compare the text-mined extractions to the structured field in the record. Whilst in some electronic health record systems this could be appropriate, in the context of this system there would be numerous issues with this. Firstly, and most importantly, the structured field in the record for occupation is not considered a ‘gold-standard’ – it is unclear how accurately employment is recorded by clinicians in the structured field, and there are only broad occupational categories available which may not accurately capture the patient’s occupational circumstances (e.g. ‘paid employment’). The structured field also cannot provide us with more information on temporality as whilst the structured field has a time/date for clinician recording, we do not have any indication of when the patient started the occupation or if they still hold that occupation. The application can also identify multiple occupations per person, whilst the structured field only provides one occupation per patient. Therefore, calculating the overlap between occupations recorded in the structured field and free-text would not be meaningful in the context of the data available in the structured field. We instead have compared the number of patients with occupations recorded in the structured field and in free-text as identified by the application, as this demonstrates the value of using text-mining to identify patient occupations (page 11, paragraph 1).

“Page 7 line 46. What is the recall rate of the developed NLP application for the extraction of the PH sections? Is there any citation for the application?”

There is no citation for the application, however following this feedback we have calculated the recall estimate for the personal history application (page 6, paragraph 3).

“Please double-check the abbreviations used in the draft. I cannot find the full names of some terms like SLaM.”

We have double checked that all abbreviations are clearly referenced in their first instance. All abbreviations were fully described in the original paper, including South London and Maudsley (SLaM) (page 6, paragraph 1).

Reviewer: 4

Reviewer Name: J. Zwaveling

“I have no comments, since the manuscript is written very clearly and of outstanding quality. 1 minor remark: reference 10 is mentioned before reference 6 and 7 in the introduction.”

This citation has been rectified.

VERSION 2 – REVIEW

REVIEWER	Mike Conway University of Utah, United States
REVIEW RETURNED	18-Oct-2020

GENERAL COMMENTS	[REVIEW] Text-mining occupations from the mental health electronic health record: a natural language processing approach using records from the Clinical Record Interactive Search (CRIS) platform in South London, UK. 18th Oct 2020 BMJ OPEN rereview ## SUMMARY I believe that the authors have largely addressed the concerns expressed in my original review. However, there remain a couple of minor points of clarification suggested below. ## COMMENTS * In my original review I suggested that you not use the term “cutting edge” as the methodology probably can’t reasonably be described as such, but it can be described as innovative in terms of the application. The text has been amended to “We have developed innovative methodology to text-mine patient occupations...”. This does not address my concern as “innovative” and “cutting edge” are — at least to my ears — near synonyms. The point I was making in my review is that the methodology can’t reasonably be described as cutting edge/innovative, though there is some innovation in the application of an established methodology to your problem (but it is not a *methodological* innovation). Suggest something like “We have developed a tool to text-mine patient occupations” instead. * “... notably in industrial research”. The term “industrial research” is ambiguous (i.e. does it mean research done *in* industrial contexts or research focused *on* industry?). Suggest something like “research on industrial injuries” * “...prove useful in forwarding our understanding...” Perhaps “addressing” is better than “forwarding” here. * In my initial review, I suggested including a reference to the Kappa statistic in NLP. I meant “reference” in the sense of “citation” (i.e. include a citation).
---

REVIEWER	Hong-Jie Dai National Kaohsiung University of Science and Technology
REVIEW RETURNED	12-Oct-2020

GENERAL COMMENTS	The authors have addressed my concerns in the revision. I just suggest the authors to carefully proofread the Page 7 line 18: using the guidelines to a gold-standard => using the guidelines to create a gold-standard
--

VERSION 2 – AUTHOR RESPONSE

We would like to sincerely thank the reviewers for taking the time to review our manuscript following edits. We are pleased that they are recommending this paper for publication. We have made the suggested changes and proof-read the paper carefully. Specifically, we have listed the changes made to the paper below in response to the reviewers:

Reviewer 2:

1. On page 13 paragraph 1: we have changed "we have developed innovative methodology" to "we have developed a tool". We understand and appreciate the distinction the reviewer is making here.
2. On page 15, paragraph 1: we have changed "industrial research" to "research on industrial injuries" to make this more specific.
3. On page 15, paragraph 2: we have changed "forwarding our understanding" to "addressing our understanding".
4. On page 10, paragraph 1: we have included a citation/reference which discusses the use of the Cohen's Kappa agreement statistic when developing a gold-standard annotated dataset for NLP (number 24).

Reviewer 3:

1. On page 7, paragraph 1: we have changed "using the guidelines to a gold-standard" to "using the guidelines to create a gold-standard".
2. We have carefully proof-read the manuscript.